# Practical Road-Resistance Functions for Expressway Work Zones in Occupied Lane Conditions

**Chi Zhang [1,2], Jihan Qin [1], Min Zhang [3,\*], Hong Zhang [1] and Yudi Hou [1]**

[1] Key Laboratory for Special Area Highway Engineering of Ministry of Education, Chang'an University, Xi'an 710064, China; zhangchi@chd.edu.cn (C.Z.); 2017121242@chd.edu.cn (J.Q.); hongzhang@chd.edu.cn (H.Z.); 2017121231@chd.edu.cn (Y.H.)

[2] China Communications Construction First Highway Consultants Co., Ltd., Xi'an 710064, China

[3] Traffic Engineering Research Institute, School of Highway, Chang'an University, Xi'an 710064, China

[*] Correspondence: minzhang@chd.edu.cn

**Abstract:** In order to create a practical road-resistance function for work zones under different lane occupation conditions, the expected speed of vehicles was calibrated in the work zone simulation model based on measured data, and simulation models were constructed for the closed half lane and the closed inside lane under different rates of trucks. Based on the statistical theory, the influence of significance of traffic volume and truck ratios for road resistance was analyzed, and a suitable truck ratio was found for the work zone. By using the optimal nonlinear fitting theory, the practical road-resistance function for work zones under different lane occupation conditions was constructed. The results showed that the road resistance is significantly affected by the traffic volume and rate of trucks. Under the same truck ratio, the road resistance linearly increases slowly when the traffic volume is less than the critical traffic volume and rapid increases irregularly when it is greater than the critical traffic volume. Under the same traffic load, the road resistance of the work zone increases with the increase in the rate of trucks, and the difference is not obvious when the traffic volume is less than the critical traffic volume, and increases gradually when it is greater than the critical traffic volume. Through the goodness of fit test and the homogeneity of variance test, the road-resistance function constructed in this paper has high goodness of fit. The practical road-resistance functions constructed in this study could be used to guide the diversion of the rebuilt/expanded highway to ensure traffic safety. Further, the study provides a theoretical basis for the construction of intelligent highway work zones.

**Keywords:** transport safety; work zone; road-resistance function; simulation; VISSIM

## 1. Introduction

In recent years, the increase in new road projects tends to be stable, whereas road reconstruction and expansion projects show an increasing trend year by year. At present, most of the reconstruction and expansion of expressways adopt the strategy of opening the border during construction, which has a great impact on the original traffic. Traffic congestion, frequent traffic accidents, and low traffic efficiency often occur. On the basis of ensuring the construction of the project, it is necessary to formulate a reasonable road network traffic organization plan and improve the traffic capacity of the regional road network. Traffic impedance is the main means to solve such traffic problems by quantifying travel time cost and providing relevant information for optimizing road network traffic organization. However, the current road resistance function is not applicable to the expressway reconstruction and expansion section, so it is necessary to construct a practical road resistance function that can adapt to the driving characteristics of the construction area.

In order to quantify the impact of traffic impedance on travelers' choice of travel path, grasp the distribution of traffic volume on urban road network, and plan reasonable urban road circulation, the research on traffic impedance has been carried out for two decades. At present, some mature road resistance function models have been established such as the United States Bureau of Highways (BPR) function, EMME/2 (INRO, Montreal, Canada)-Tapered delay function, logit delay function, Akcelik delay function, and generalized cost function based on the BPR delay curve [1–3]. On the basis of comprehensive consideration of traffic flow, traffic environment, traffic organization and other factors, most scholars have re-calibrated, revised, and improved the existing road resistance function model [4–12]. Based on the measured data or simulation data, some scholars have studied and compared the road resistance functions under different conditions [12–15]. Some scholars have re-deduced the road resistance function from the aspects of traffic flow and traveler's comprehensive cost, and constructed a new or composite road resistance function model [16,17].

However, the existing road-resistance function is for urban roads and normal expressways, and it cannot be applied to the work zone area of expressways. The current road-resistance function does not take into account the driving characteristics of construction areas and the lanes occupied in the work zone, and there is a large difference between the load resistance and the real construction area. Furthermore, the construction of the road-resistance function will directly affect the traffic assignment on the regional road network for reconstruction and expansion and the driving conditions through the reconstruction and expansion project. Therefore, it is necessary to study the road-resistance function of the reconstruction and expansion and to consider the driving characteristics of the construction area before building a practical road-resistance function. This paper calibrates the simulation model based on the actual measurement data of the construction area and conducts research on the practical road-resistance functions of work zone under different traffic conditions. The results show that the proposed models have better adaptability than the conventional model when the load ratio ($q/c$) is relatively high.

## 2. Road-Resistance Model

At present, the most widely used internationally is the BPR road-resistance function model. Although this model is not consistent with the state of the expressway in China, its expression in the road-resistance function is more accurate. Moreover, its mathematical properties are useful regarding monotonicity and conductibility, so this paper uses the BPR function as a basis to build a road-resistance function model for the construction area. Its mathematical form is expressed in Equation (1).

$$T(q) = \frac{T_0}{\left(1 + \alpha \left(\frac{q}{c}\right)^\beta\right)} \tag{1}$$

$T(q)$: travel time on the road when traffic volume is $q$;
$T_0$: zero flow rate, vehicle travel time on the road section;
$c$: Design capacity of the road section;
$\alpha, \beta$: Model parameters.

Among them, parameter values of $\alpha = 0.15$, $\beta = 4$ are called the classical BPR model, which is based on the velocity-flow curve of the 1965 HCM (Highway Capacity Manual) [18]. Later, another study revised the classical BPR model. The revised parameter was $\alpha = 1$, $\beta = 10$. This function has the same velocity value at $v/c = 1$ as in the 1985 HCM (free flow velocity 30 mph).

## 3. Expected Speed Calibration in the Construction Area

It is necessary to carry out reasonable traffic assignment strategy before design the traffic management of the work zone. Therefore, in this paper, simulation experiments are performed on the construction area considering the driving characteristics of the work zone. This paper first calibrates

the simulation model of the work zone to build a universal road-resistance function, which can ensure the simulation data is sufficiently authentic.

The actual data collection site for the construction area is a specific work zone of the Anhui expressway, which is a two-way four-lane expressway with a design speed of 120 km/h. The speed limit for the work zone was 60 km/h. The data collection time was 13–14 September 2017. The primary collection objects were passenger cars and trucks, and the collected data types were traffic volume and location speed [19,20]. The traffic volume was selected using a manual counting method, and the entire camera is taken at the same time for the next data check. The collection device for the vehicle speed is a handheld radar speedometer with an error range of ±2 km/h, and the accuracy meets the test requirements. The velocity profile is shown in Figure 1. The weather conditions during the test data collection are on the right, and the data comes from free traffic flow.

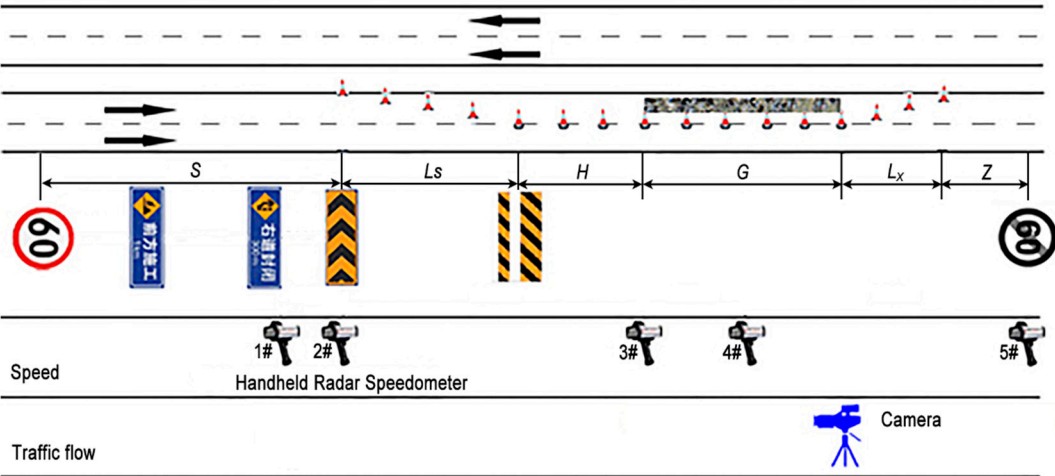

**Figure 1.** The measured sections and field situations. *S* denotes the warning area, *Ls* denotes the upstream transition area, *H* denotes the buffer area, *G* denotes the work area, *L$_X$* denotes the downstream transition area, *Z* denotes the termination area.

### 3.1. Data Processing

Because of the large error in the individual values of the collected data, this article uses a single-sample K-S to determine that the acquisition rate obeys the normal distribution and is filtered according to the 3σ criteria. The formula is as follows (2):

$$P(|x - \mu| > 3\sigma) \leq 0.003 \tag{2}$$

In Equation (2), *μ* and *σ* represent the mathematical expectation and standard deviation of the normal distribution, respectively. It can be seen that the probability of data values greater than $\mu + 3\sigma$ or less than $\mu - 3\sigma$ in the data values was small. Therefore, a measured value that deviated from the average value by more than three times the standard deviation was considered to be an abnormal value and was rejected.

### 3.2. Desired Speed Calibration

This paper proposes a microscopic simulation model calibration method that aims at optimizing a measure-of-effectiveness (MOE) distribution. For the parameters of behavioral models that affect these indicators through sensitivity tests, parameter calibration is performed through cross-experimental design or heuristic algorithm [15].

When carrying out the work zone simulation, not only should we pay attention to the calibration of the operating speed, but also to the relevant queue length, delay time, traffic collisions, etc. It can

bring the simulation closer to the actual value in many aspects. This paper considers the primary goal of optimizing MOE distribution and proposes a systematic calibration method:

Step 1: Collect the necessary data for traffic simulation model, including road geometry data (CAD or work zone layout) and traffic flow data (traffic flow in construction area, traffic flow ratio, vehicle speed distribution, headway spacing distribution, etc.).

Step 2: Use traffic simulation software to model the construction area based on the collected data.

Step 3: Select the construction area indicator MOE. This study considers the convenience and accuracy of video data acquisition and selects the travel speed as the model-calibrated MOE from the traditional efficiency indicators.

Step 4: Perform sensitivity tests on the set of driving behavior model parameters in the simulation software. Through the variance test, determine the set of parameters that have a significant effect on the efficiency index.

Step 5: Set the fitness function to the GEH (Proposed by Geoffrey E. Havers to compare the matching degree of two sets of traffic data) index [15] (Formula (3)). Use a genetic algorithm to calibrate the sensitive parameter set. In the aspect of genetic algorithm strategy selection, this paper selects commonly used strategies and adopts individual random sampling (namely, a random set of parameters for each parameter is randomly assigned within a specific range, which is called a random individual) and is coded. Use the two-player competition operator to select "good" individuals. Select the uniform crossover strategy (0.5) to perform the "crossover" operation on the remaining high-quality individuals, generating new individuals, selecting the uniform mutation strategy (0.5), and mutate the individual until the algorithm converges.

$$GEH = \sqrt{\frac{(v_2 - v_1)^2}{0.5(v_1 + v_2)}} \tag{3}$$

Step 6: To measure the closeness of the distribution of the simulated MOE and the actual MOE, this paper selects the Kullback–Leibler divergence (KL-D) as a fitness function [16] (Equation (4)):

$$D_{KL}(P\|Q) = \int_{-\infty}^{\infty} p(x) \log \frac{p(x)}{q(x)} dx \tag{4}$$

where $p(x)$ and $q(x)$ are the probability density functions of the simulated and actual observed distributions, respectively.

The critical parameters of driving behavior in the VISSIM (PTV Group, Karlsruhe, German) simulation software were calibrated according to the average queue length at the intersection of video observations. Through sensitivity tests, it was found that the three parameters in the car-following model significantly influence the efficiency index, including the average parking distance, additional portion of the safety distance, and multiple of the safety distance. The genetic algorithm was used to calibrate the optimal values of these three parameters (Table 1).

**Table 1.** Default and calibration values of VISSIM simulation parameters.

| VISSIM Parameters | Default Value | Calibration Value |
|---|---|---|
| Average parking distance | 2.1 | 2.0 |
| Additional distance to safety distance | 2.5 | 2.4 |
| Multiplier part of safety distance | 3.2 | 3.1 |

After inspection, the precision of the simulation speed could be obtained.

## 4. Practical Road-Resistance Function in the Construction Area

### 4.1. Simulation Model Construction

This study uses VISSIM microscopic simulation software for traffic simulation. The satellite map of the field test section was imported into VISSIM as a base map, and a primary road section was drawn on it. The layout of the construction section is drawn according to Table 2 and Figure 1. Each construction section is drawn using the "links" button. It should be noted that when there is a case of segment connection, "connectors" is used for the setting, such as the above and downstream transition zone. The type of model is "freeway," and the number of lanes is set to two narrow lanes. Each lane is 3.75 m wide. The vertical slope of the road is set according to the actual situation on the road.

**Table 2.** Layout parameters of the work zone.

| Condition | S | $L_S$ | H | G | $L_X$ | Z |
|---|---|---|---|---|---|---|
| Closed inside lane | 2000 | 120 | 80 | 3000 | 120 | 30 |
| Closed half lane | 2000 | 120 (200) | 80 | 3000 | 120 | 30 |

The closed inside lane is a traffic organization form of the construction area, which merges vehicles into outer lane and does not change the direction of traffic flow, as shown in Figure 2.

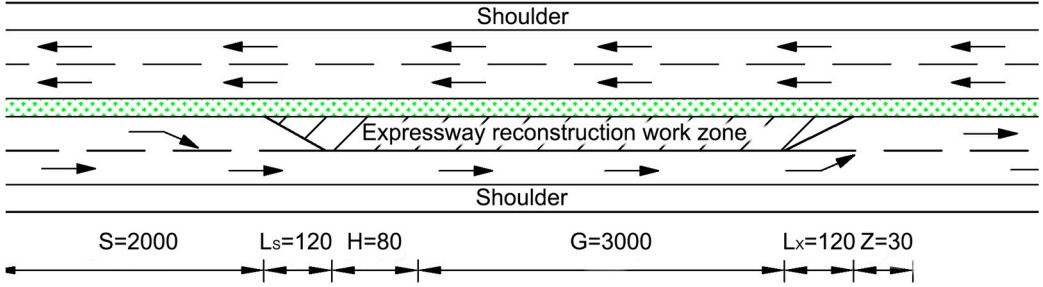

**Figure 2.** Schematic diagram of the closed inside lane.

The closed half lane is a traffic organization form of the construction area, which transfers the traffic flow to the opposite lane through the opening of the medial divider, as shown in Figure 3.

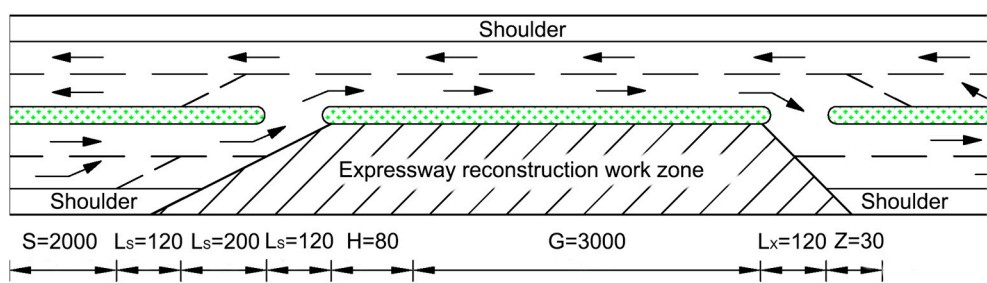

**Figure 3.** Schematic diagram of the closed half lane.

### 4.2. Determination of Sample Size

To ensure the reliability of the model calibration, some simulation data is needed. The ideal sample size is determined using Equation (5):

$$N \geq \left( \frac{S \times m}{E} \right)^2 \left( 1 + \frac{R^2}{2} \right) \tag{5}$$

Including the *N*-sample volume, *S* is the standard deviation of sample data of observation data, where $S = 8.5$ km/h in two lanes, $S = 6.8$ km/h in four lanes, and $S = 5.2$ km/h in six lanes. The M-normal distribution of the upper probability number when the confidence is 95% (90%) was $m = 1.96$ (1.65), and *E* is the run error, assuming a speed tolerance of 2–5 km/h. The value of *R* satisfies Equation (6).

$$\text{R-constant} = \begin{cases} 0.00, \text{Average speed} \\ 1.04, \text{15th or 85th percentile average speed} \\ 1.64, \text{5th or 95th percentile average speed} \end{cases} \tag{6}$$

The object of this study was a two-way, four-lane expressway, and the minimum sample calculation is used as an example. The allowable error was 2 km/h, and this calibration data uses the 85th-percentile average speed, so $R = 1.04$, and the confidence level was 95%, so $m = 1.96$. Therefore, the minimum number of samples used to calibrate the road-resistance function was 69.

Because the traffic volume and vehicle type composition were relatively stable in the measured data, the study of the impact of different traffic volumes and different types of vehicles on the road-resistance function had greater limitations. To ensure the diversity of data, this paper used VISSIM simulation software to construct the driving environment under different traffic conditions and different rate of heavy vehicle. In the selection of parameters, the traffic volume was 1950 pcu/h and the minimum traffic volume was 750 pcu/h. With 50 steps, the number of experimental groups was expanded. Therefore, there were 25 simulation experiments under the conditions of each specific rate of heavy vehicle. In this simulation experiment, [2.5%, 32.5%] was divided into equal distances with 2.5% of the ratio of heavy vehicles as the change step, and the simulation experiments are carried out on the proportion of these 13 groups of vehicles.

Sample sizes of different truck ratios under closed half lane and closed inside lane conditions are shown in Table 3.

**Table 3.** Sample size under different truck proportions.

| The Way of Closed Lane | The Rate of the Heavy Vehicle | Sample Sizes |
|---|---|---|
| Closed inside lane | $P \leq 10\%$ | 100 |
| | $10\% < P \leq 25\%$ | 150 |
| | $P > 25\%$ | 75 |
| Closed half lane | $P \leq 7.5\%$ | 75 |
| | $7.5\% < P \leq 17.5\%$ | 100 |
| | $17.5\% < P \leq 25\%$ | 75 |
| | $P > 25\%$ | 75 |

It can be seen that each group of tests meets the requirement of minimum sample size 69.

### 4.3. Significance of Influencing Factors

In this simulation experiment, the proportion of models was divided into 13 groups, the traffic volume was divided into 25 groups, and 325 sets of data were simulated under the conditions of closed half-full lanes and enclosed inside lanes, and the travel time of the sections under different conditions was obtained. The simulation results are plotted on a three-dimensional surface, as shown in Figures 4 and 5.

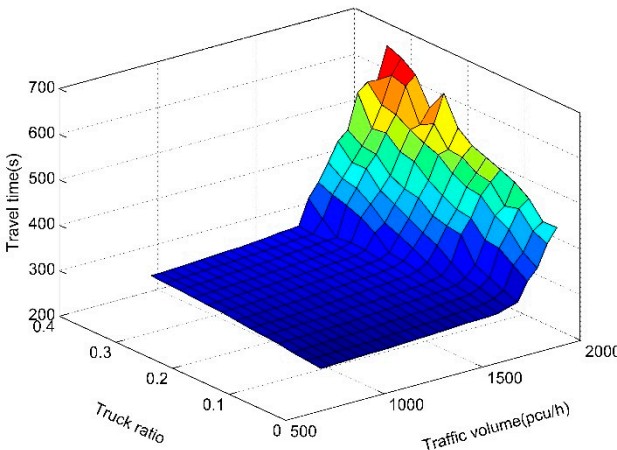

**Figure 4.** Surface map of the travel time under the condition of the closed inside lane.

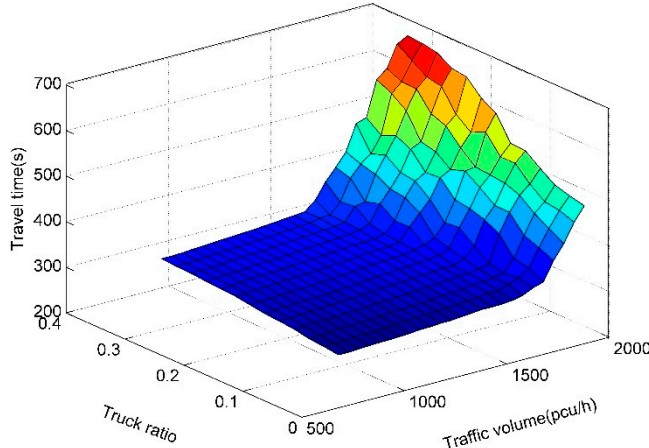

**Figure 5.** Surface map of the travel time under the condition of the closed half lane.

From Figures 4 and 5, the changes in the travel time are roughly the same in both methods of closing the half-full lane and closing the inside lane. Although the traffic volume and the rate of heavy vehicles were gradually increasing, the change in driving time had the same tendency. Before the critical traffic volume, the trip time increased linearly with the increase of the traffic volume and the rate of heavy vehicles. However, after the critical traffic volume, the travel time increased and the regularity deteriorated; that is, the randomness of the travel time under congestion conditions became stronger. The value of critical traffic volume was greatly affected by the rate of heavy vehicles. Under the condition of closed inside lanes, the critical traffic volume was about 1700 pcu/h in the case of a 10% heavy-vehicle ratio. In the 20% case, the critical traffic volume was about 1650 pcu/h, and in the 30% case, the critical traffic volume was about 1550 pcu/h. That is, as the rate of heavy vehicles increased, the critical traffic volume also decreased. According to this analysis, the average critical traffic volume was 1650 pcu/h. At this time, the expressway was a third-level service, and after the critical traffic volume was greater than the critical traffic volume, it entered a congested state, which is a fourth-level service. After the critical traffic volume, the traffic volume continued to increase, and the travel time increased significantly. At the same time, the rate of heavy vehicles had a significant influence on the travel time. Therefore, after the critical traffic volume was reached, the effect of different vehicle types on traffic diversion was significant.

Before the critical traffic volume, the travel time of the vehicle slowly increased. After the critical traffic volume, the travel time increased rapidly. Using the difference trend as an indicator, the critical traffic under different vehicle types was studied. That is, under the condition of the same heavy vehicle

ratio, the critical traffic volumes under the conditions of the closed half-width lane and the closed inside lane were determined, as shown in the Table 4.

**Table 4.** Critical traffic calculation table.

| Rate of Heavy Vehicles | 2.5% | 5% | 7.5% | 10% | 12.5% | 15% | 17.5% | 20% | 22.5% | 25% | 27.5% | 30% | 32.5% |
|---|---|---|---|---|---|---|---|---|---|---|---|---|---|
| Closed Half Lane | 1800 | 1750 | 1750 | 1750 | 1700 | 1650 | 1650 | 1600 | 1600 | 1600 | 1550 | 1550 | 1500 |
| Closed Inside Lane | 1800 | 1750 | 1750 | 1700 | 1700 | 1650 | 1650 | 1650 | 1600 | 1600 | 1600 | 1550 | 1550 |

It can be seen that the critical traffic volume for closing the inside or half-width lane did not differ much (Table 3). In other words, the critical traffic volume was not affected by the closure method but was mainly affected by the rate of heavy vehicles. With the increase in the rate of heavy vehicles, the critical traffic volume gradually increased. Because this study used 50 traffic steps, only a rough estimate of the critical traffic volume could be obtained, which was not accurate. This critical traffic volume can provide suggestions for the essential diversion strategy of the work zone with a speed limit of 60 km/h.

Based on the application of two-way analysis of variance, this paper examined the impact of traffic flow and the rate of heavy vehicle on travel time statistically. It was assumed that the traffic was factor A that influenced the travel time and the vehicle proportion was regarded as the travel time influence factor B. Here, we assumed that the traffic flow and the rate of heavy vehicles were independent of each other and there was no correlation between them. In this paper, 325 sets of simulation data under closed half-lane conditions were used for two-factor analysis with confidence coefficient $\alpha = 0.05$.

According to calculations, it could be concluded that $F_A = 33.474$ was much larger than the critical value of 1.75, and $F_B = 121.590$ was far greater than the critical value of 1.52 (Table 5). Therefore, both A and B factors should accept the assumption that the flow rate and rate of heavy vehicles had a significant impact on the travel time in the work zone.

**Table 5.** Two-factor analysis of variance for the closed half lane.

| Difference Source | Sum of Squares of Deviations | Freedom | Mean Square Deviation | F | Critical Value |
|---|---|---|---|---|---|
| A | 269,895.5659 | 12 | 22,491.29716 | 33.474 | 1.75 |
| B | 1,960,692.807 | 24 | 81,695.53363 | 121.590 | 1.52 |
| Errors | 193,506.129 | 288 | 671.8962814 | | |
| Total | 2,424,094.502 | 324 | | | |

*4.4. Reasonable Interval of the Rate of Heavy Vehicles*

From the above significant test, it can be seen that both the traffic volume and the rate of heavy vehicle had a significant effect on the travel time in the work zone. In the study of the road-resistance function of urban roads, the division of interval for heavy vehicles was at dividing lines of 10%, 20%, and 30% [21]. However, there is no specific study on the determination of the dividing line. Using the trend difference as a research indicator, the division of interval of the rate of heavy vehicle in the work zone of the highway construction area was studied. The trend difference calculation formula is given as Equation (7):

$$T_j = \sum_{i=1}^{n} \left| t_{i+50, j} - t_{i,j} \right| \tag{7}$$

where

$T_j$: trend difference in the ratio of *j* to *j*;

$t_{i,j}$: the proportion of vehicles is *j*, and the traffic volume is the travel time under *i* simulation conditions;

*j*: the rate of heavy vehicles;

*i*: road resistance simulated traffic volume.

The Figure 6 is obtained by calculating the trend difference under the two traffic conditions of the enclosed half lane and closed inside lane.

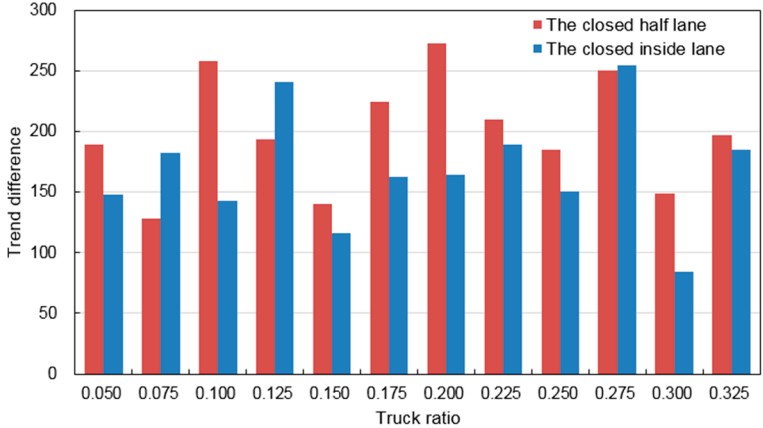

**Figure 6.** Trend difference in the closed half lane (red bars) and the closed inside lane (blue bars).

Our results showed that the trend difference in the closed half lane shows a fluctuation such that the minimum difference was 5–7.5%, the maximum difference was 17.5–20%, and the trend of the model proportions of 7.5% and 5% were the closest to no difference (Figure 4). The difference in trend between the 17.5% and 20% models was the largest. In this study, 250 was used as the critical value of the trend difference. It can be seen that there were three peaks of 10%, 20%, and 27.5% in the model, but the three peak positions were not the criteria for vehicle classification. When the rate of trucks was 10%, there was a crest, indicating that the difference between the 10% and 7.5% cases was large, and it was considered that these two model ratios should have different road-resistance function values. Taking the trend difference of the three wave peaks as the basis for the interval of the rate of trucks, the reasonable interval of the truck ratio in the work zone was divided into $P \leq 7.5\%$, $7.5\% < P \leq 17.5\%$, $17.5\% < P \leq 25\%$, and $P > 25\%$.

Our results also showed the trend difference under closed inside lane conditions was with a minimum difference of 27.5–30% and a maximum difference of 25–27.5%. The trend of the model proportions of 27.5% and 30% was the closest, with 25% and 27.5% of models having the most significant trend difference. The critical value of the trend difference is 200 under the closed inside lane condition. It can be seen that there were 12.5% and 27.5% peaks in the model proportion. According to the same division method, the model ratio could be divided into $P \leq 10\%$, $10\% < P \leq 25\%$, and $P > 25\%$.

### 4.5. Road-Resistance Functions for Expressway Work Zones

For the BRP function, $T$, $T_0$, $q$, $c$ can be obtained from simulation. At this time, the calibration of the BRP function in the construction area is transformed into the calibration of the parameters $\alpha, \beta$. The calibration of these parameters can transform the original formula into a parameter-calibration problem.

Set $y = \frac{T(q)}{T_0}$, $x = \frac{q}{c}$.

The formula is transformed into $y = 1 + \alpha x^\beta$, $y - 1 = \alpha x^\beta$, and the logarithm of both sides is $\ln(y - 1) = \beta \ln \alpha + \beta \ln x$.

Set $\ln(y - 1) = y_0$, $\ln x = x_0$, $A = \beta \ln \alpha, B = \beta$.

It was determined that the calibration of $\alpha$ and $\beta$ was converted to the parameter-fitting problem of $A$ and $B$.

*A* curve-fitting model based on Chebyshev's meaning in the least-squares method was used.

Take the example of closed inside lane in which $P \leq 10\%$ for an introduction and other results were shown in Table 6. Using the universal global optimization method and the McCourt method in 1stopt,

the model parameters $\alpha, \beta$ were calibrated. After entering the parameter into 1stopt, after 29 iterations, the model reaches the convergence criterion.

**Table 6.** Practical road-resistance function of expressway reconstruction work zone.

| The Way of Closed Lane | The Rate of the Heavy Vehicle | Road-Resistance Function | Number of Iterations | F | $F_{0.05}(n)$ | $\chi^2$ | $\chi^2{}_{0.05}(n)$ | $R^2$ | Goodness of Fit |
|---|---|---|---|---|---|---|---|---|---|
| Closed inside lane | $P \leq 10\%$ | $T = T_0\left(1 + 1.429\left(\frac{q}{c}\right)^{4.923}\right)$ | 29 | 396.622 | 1.35 | 0.487 | 129.070 | 0.802 | good |
| | $10\% < P \leq 25\%$ | $T = T_0\left(1 + 1.897\left(\frac{q}{c}\right)^{4.086}\right)$ | 24 | 880.894 | 1.22 | 0.975 | 185.312 | 0.857 | good |
| | $P > 25\%$ | $T = T_0\left(1 + 2.674\left(\frac{q}{c}\right)^{4.202}\right)$ | 16 | 697.786 | 1.53 | 0.505 | 100.345 | 0.909 | good |
| Closed half lane | $P \leq 7.5\%$ | $T = T_0\left(1 + 1.140\left(\frac{q}{c}\right)^{3.823}\right)$ | 19 | 296.343 | 1.53 | 0.283 | 100.345 | 0.802 | good |
| | $7.5\% < P \leq 17.5\%$ | $T = T_0\left(1 + 1.500\left(\frac{q}{c}\right)^{3.634}\right)$ | 18 | 427.711 | 1.35 | 0.600 | 129.070 | 0.814 | good |
| | $17.5\% < P \leq 25\%$ | $T = T_0\left(1 + 1.961\left(\frac{q}{c}\right)^{3.657}\right)$ | 28 | 518.313 | 1.53 | 0.503 | 100.345 | 0.877 | good |
| | $P > 25\%$ | $T = T_0\left(1 + 2.431\left(\frac{q}{c}\right)^{3.797}\right)$ | 24 | 780.189 | 1.53 | 0.536 | 100.345 | 0.914 | good |

This article uses the mathematical optimization software 1stopt developed by the seven-dimensional high-tech company. Through its unique global optimization algorithm, the optimal solution is finally found as Equation (8).

$$\alpha = 1.429, \beta = 4.923 \tag{8}$$

In the condition of the closed inside lane, the road-resistance function model under the condition that the ratio of trucks is less than 10% is given as Equation (9):

$$T = T_0\left(1 + 1.429\left(\frac{q}{c}\right)^{4.923}\right) \tag{9}$$

The homogeneity of variance test (F-test) value of the fitted model was F = 396.622. The sample data were used and there were model variables. The critical value of the F test was $F_{0.05}(100,100) = 1.35$. It was evident that the F-value was far more significant than the critical value. Because $\chi^2(100)$ could not be derived from the look-up table, this paper used Equation (10) to perform the calculations. $\chi^2{}_{0.05}(100) = 100.345$, which was much greater than $\chi^2 = 0.487$. Therefore, the goodness-of-fit test results show that the fitting results to the sample data were very good.

$$\chi^2{}_p(n) = \frac{1}{2}\left(\mu_p + \sqrt{2n-1}\right)^2 \tag{10}$$

From the Figure 7, the proposed models and the conventional model began to differ when $q/c$ equaled 0.3 in the closed inside lane. The proposed models began to differ with different truck ratios when $q/c$ equaled 0.47 and the difference became larger and larger with the increase of $q/c$.

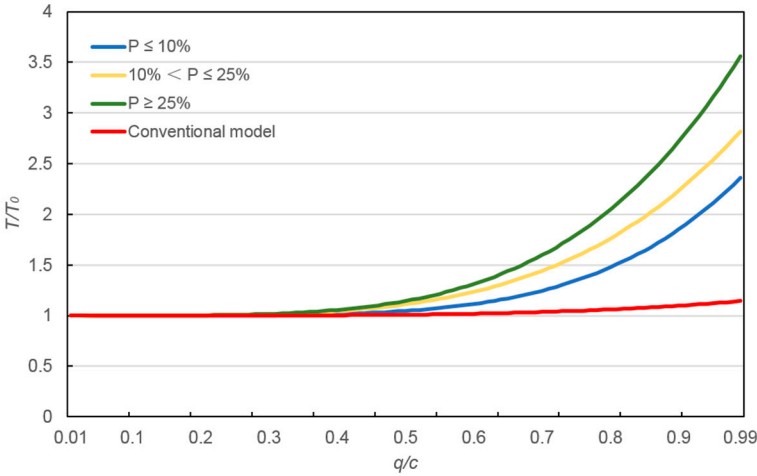

**Figure 7.** Comparison between the proposed models and the conventional model (red line) in the closed inside lane.

From the Figure 8, the proposed models and the conventional model began to differ when $q/c$ equaled 0.25 in the closed half lane. The proposed models began to differ with different truck ratios when $q/c$ equaled 0.39 and the difference became larger and larger with the increase of $q/c$.

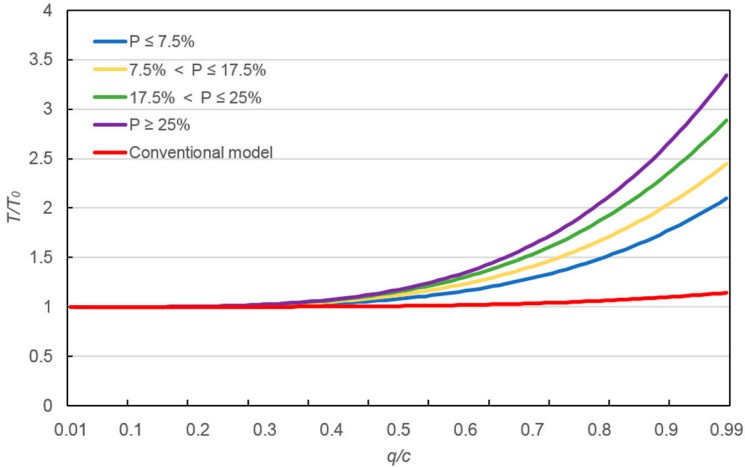

**Figure 8.** Comparison between the proposed models and the conventional model (red line) in the closed half lane.

It can be seen that the conventional models were very close to the proposed model when $q/c$ was small. Furthermore, the road-resistance function in different truck ratios conditions differed greatly when $q/c$ was large. Therefore, when $q/c$ was large, the proposed models have better adaptability than the conventional model.

## 5. Conclusions

Our study showed that the truck ratio and traffic volume had a significant impact on travel time. Before the VISSIM simulation for travel time, we calibrated the expected speed based on the actual measurement data in the expressway work zone. The study provides a simulation way for calibrating the expected speed using measurement data, which could be applied to calibrate other parameters of the traffic safety field. In addition, other factors affecting road resistance of the highway work zones need further study.

Our study found that the rate of trucks in the closed inside lane was divided into $P \leq 10\%$, $10\% < P \leq 25\%$, and $P > 25\%$. The rate of trucks in the closed half lanes aws divided into $P \leq 7.5\%$, $7.5\% < P \leq 17.5\%$, $17.5\% < P \leq 25\%$, and $P > 25\%$. A universal global optimization algorithm was used to calibrate the road-resistance function under different traffic conditions and road-closure methods. Both the F test and goodness-of-fit test demonstrated a good fit. We are now working on the road-resistance functions under different speed limitations, aiming to thoroughly study the vehicle operating conditions and provide guidance for traffic safety in highway work zones.

The study provides a method for constructing practical road-resistance functions for expressway work zone, which could be used to guide the diversion of the rebuilt/expanded highway to ensure the traffic safety. Further, the study provides a theoretical basis for the traffic assignment of networks and traffic-management plans in the rebuilt/expanded highway, and a reference for construction of intelligent highway work zones.

**Author Contributions:** Conceptualization, C.Z. and M.Z.; Methodology, C.Z. and J.Q.; Software, H.Z. and J.Q.; Validation, Y.H. and H.Z.; Formal Analysis, C.Z. and J.Q.; Investigation, M.Z. and Y.H.; Resources, C.Z.; Data Curation, M.Z.; Writing—Original Draft Preparation, J.Q. and H.Z.; Writing—Review and Editing, C.Z. and M.Z.; Funding Acquisition, C.Z.

**Funding:** This research was funded by National Key R & D project [2017YFC0803906], the Postdoctoral Fund of Ministry of Education of China [2016M590915], Shaanxi Natural Science Basic Research Project [S2017-ZRJJ-MS-0603].

**Acknowledgments:** The authors appreciate Key Laboratory for Special Area Highway Engineering of Ministry of Education for providing software and Anhui Transportation Holding Group CO., LTD. for providing basic information.

**Conflicts of Interest:** The authors declare no conflict of interest.

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
