# Peer review of "Practical Road-Resistance Functions for Expressway Work Zones in Occupied Lane Conditions"

_sustainability, doi:10.3390/su11020382_

Round 1
Reviewer 1 Report
The paper proposed road-resistance functions based on simulation model for expressway work zone. However, the authors need to address the following concerns.
1) Authors should add some broader contexts and importance of road-resistance about your research in the introduction.
2) Previous research is needed to be summarized in a form of table with appropriate categories.
3) The definitions of “the closed inside lane” and “the closed half lane” are unclear. Please visualize them in Fig 1 or use another figure.
4) The authors show the experiment results according to the different traffic conditions and rates of heavy vehicle using the proposed models. However, the performance comparisons with existing models should be added to show the numerical improvements of the suggested concepts.
5) Please summarize the novelty of the proposed models compared to the conventional models in introduction session.
6) The background and the needs regarding to this study should be introduced in the beginning of the introduction. The current version has too much summarized. And, the first paragraph in introduction is too long.
Author Response
Response to Reviewer 1 Comments
Point 1: Authors should add some broader contexts and importance of road-resistance about your research in the introduction.

Response 1: Considering the reviewers’ suggestion, we have added some broader contexts and importance of road-resistance in paragraph 1.
Point 2: Previous research is needed to be summarized in a form of table with appropriate categories.
Response 2: Considering the reviewers’ suggestion, we have optimized the logic of the research review and refined the structure of the introduction.
Point 3: The definitions of “the closed inside lane” and “the closed half lane” are unclear. Please visualize them in Fig 1 or use another figure.
Response 3: Considering that there are two forms of traffic organization in this paper which are closed inside lane and closed half lane, we can not explain both them in figure 1, so we have added figure 2 and figure 3 to explain the definitions of “the closed inside lane” and “the closed half lane”.
Point 4: The authors show the experiment results according to the different traffic conditions and rates of heavy vehicle using the proposed models. However, the performance comparisons with existing models should be added to show the numerical improvements of the suggested concepts.
Response 4: Considering the reviewers’ suggestion, we have added the comparison between the proposed models and the conventional model at the end of the paper, and explained the advantages of the proposed models.
Point 5: Please summarize the novelty of the proposed models compared to the conventional models in introduction session.
Response 5: It is really valuable as reviewers suggested that the introduction session needs to summarize the novelty of the proposed models compared to the conventional models. In the third paragraph of the introduction, we have added a description of the novelty of the proposed models compared with the conventional model.
Point 6: The background and the needs regarding to this study should be introduced in the beginning of the introduction. The current version has too much summarized. And, the first paragraph in introduction is too long.
Response 6: Considering the reviewers’ suggestion, we have optimized the logical structure of the first paragraph of the introduction, highlighted the background and necessity of this study. And the research summary has been summarized and simplified.
Reviewer 2 Report
The research is well-structured but there are some grammatical errors.
Greater emphasis should be given to the case study analyzed by inserting a geolocation or images of the area.
In figure 1 it is better to add the length as a value instead in table 2 it is better to put a small legend that recalls the parameters together with the units of measurement.
From line 199 to 207 it is confusing to read data, therefore, it is considered more appropriate to insert a table or a bulleted list. A greater explanation for figure 4 must be inserted.
Author Response
Response to Reviewer 2 Comments
Point 1: The research is well-structured but there are some grammatical errors.
Response 1: We are so sorry that the linguistic imperfections, although we have tried our best to avoid it. We have perfected the linguistic of the text and corrected grammatical errors through the revision of an expert in English.
Point 2: Greater emphasis should be given to the case study analyzed by inserting a geolocation or images of the area.
Response 2: In this paper, only when the speed of the construction area is calibrated, it is related to the reconstruction and expansion of an expressway in Anhui Province. But the calibration of the practical road resistance function model in the construction area is based on the simulation data of Vissim micro-simulation software, so we add Vissim simulation sketch (Figure 2, Figure 3) to illustrate.
Point 3: In figure 1 it is better to add the length as a value instead in table 2 it is better to put a small legend that recalls the parameters together with the units of measurement.
Response 3: Considering that there are two forms of traffic organization in this paper which are closed inside lane and closed half lane, we can not label both them in figure 1, but we have added figure 2 and figure 3 to label and explain the parameters in Table 2.
Point 4: From line 199 to 207 it is confusing to read data, therefore, it is considered more appropriate to insert a table or a bulleted list. A greater explanation for figure 4 must be inserted.
Response 4: Considering the reviewers’ suggestion, we have classified and explained the data by inserting tables (Table 3). We determine the segment of truck ratio by the maximum and minimum of trend difference between two traffic organization modes. Figure 4 is to better indicate the peak value of trend difference.
